# Catheter-Associated Urinary Tract Infections: Understanding the Interplay Between Bacterial Biofilm and Antimicrobial Resistance

**DOI:** 10.3390/ijms26189193

**Published:** 2025-09-20

**Authors:** Desiye Tesfaye Tegegne, Iain J. Abbott, Błażej Poźniak

**Affiliations:** 1Department of Pharmacology and Toxicology, Faculty of Veterinary Medicine, Wroclaw University of Environmental and Life Sciences, ul. Norwida 31, 50-375 Wrocław, Poland; desiye.tegegne@upwr.edu.pl; 2Animal Biotechnology Research Program, National Agricultural Biotechnology Research Center, Ethiopian Institute of Agricultural Research, Holeta P.O. Box 249, Ethiopia; 3Department of Infectious Diseases, The Alfred Hospital and School of Translational Medicine, Monash University, Melbourne, VIC 3004, Australia; iain.abbott@monash.edu; 4Microbiology Unit, Alfred Health, Melbourne, VIC 3004, Australia; 5Centre to Impact AMR, Monash University, Melbourne, VIC 3004, Australia

**Keywords:** antimicrobial resistance, bacterial biofilm, CAUTIs, an in vitro dynamic model, urinary catheter

## Abstract

The increasing use of urinary catheters in healthcare, driven by an aging population and escalating antimicrobial resistance, presents both benefits and challenges. While they are essential to managing urinary retention and enabling precise urine output monitoring, their use significantly increases the risk of catheter-associated urinary tract infections (CAUTIs), the most common type of healthcare-associated infection. CAUTI risk is closely linked to the duration of catheterization and the formation of bacterial biofilms on catheter surfaces. These biofilms, often composed of polymicrobial communities encased in an extracellular matrix, promote persistent infections that are highly resistant to conventional antimicrobial therapies. Common CAUTI uropathogens include *E. coli*, *E. faecalis*, *P. aeruginosa*, *P. mirabilis*, *K. pneumoniae*, *S. aureus*, and *Candida* spp. The complexity and resilience of these biofilm-associated infections underscore the urgent need for innovative treatment strategies. Therefore, dynamic in vitro bladder infection models, which replicate physiological conditions such as urine flow and bladder voiding, have become valuable tools for studying microbial behavior, biofilm development, and therapeutic interventions under real clinical conditions. This review provides an overview of CAUTIs, explores the role of biofilms in sub-optimal responses to antimicrobial treatment and advances in model systems, and presents promising new approaches to combating these infections.

## 1. Introduction

Over recent decades, medical devices have played an increasingly important role in healthcare services [1]. They support the diagnosis, prevention, and treatment of a variety of medical conditions, making them a crucial component of healthcare systems [2]. An aging population and improved access to advanced healthcare are causing an increase in the demand for urinary catheters (UCs) [3]. However, UCs are prone to bacterial colonization, leading to clinical infections, and are major contributors to the increasing occurrence of healthcare-associated infections (HAIs) [4,5]. According to the WHO, approximately 5 million HAIs occur in acute care hospitals in Europe annually, leading to 50,000 deaths and additional costs ranging from EUR 13 to 24 billion annually [6].

UCs are commonly used indwelling medical devices used to drain urine from the bladder and applied in many different medical conditions, including urinary retention and incontinence, anatomical and neurodegenerative disorders, in critically ill patients in the intensive care unit, and as part of post-operative management [7,8]. Different UC types are available, with each type serving distinct purposes, such as single-use, intermittent, short-term, and long-term (>28 days, spanning months to years) indwelling catheters [9,10]. UCs are most commonly employed in patients for short-term catheterization (<14 days) [11], rather that long-term catheterization (>28 days, spanning months to years) [12]. Hospitals adhere to different management systems for the insertion and removal of UCs. Nevertheless, despite the implementation of catheter-associated urinary tract infection (CAUTI) prevention bundles and care measures, UCs remain vulnerable to bacterial colonization and subsequent infection [13,14,15]. Moreover, UCs are continuously or sporadically flushed with urine, a warm and electrolyte-rich medium [16], and this stimulates the formation of biofilm, which increases the risk of CAUTIs [16].

Accounting for 40% of HAIs globally, CAUTIs are among the most common infections associated with indwelling medical devices and significantly contribute to increased morbidity and mortality in hospitalized patients [17,18]. The likelihood of experiencing CAUTIs is linked to the duration of catheterization, with the daily increase in the risk of bacteriuria being between 3 and 10%, depending on the study [17,19,20]. Approximately, 80% of hospital-associated UTIs are linked to indwelling UC usage [21]. In the US, it is estimated that more than 30 million urinary catheters are inserted annually, which has been associated with an exponential rise in the incidence of CAUTIs [22]. Furthermore, it is predicted that the annual cost of preventable CAUTIs ranges from USD 115 million to USD 1.82 billion [23]. Therefore, the global burden of CAUTIs is related to significant financial, social, and medical costs [24].

It is estimated that biofilms are responsible for more than 80% of chronic and persistent microbial infections in humans and animals [25,26]. Biofilms have a significant role in the pathogenesis of CAUTIs [27]. CAUTIs are associated with localized effects with biofilm build-up on the catheter and encrustation, development of cystitis and pyelonephritis, and also systemic complications leading to bacteremia and septic shock [28]. Polymicrobial infections including both Gram-positive and Gram-negative bacterial species are commonly responsible for CAUTIs, with the most pathogens reported being *Escherichia coli*, *Enterococcus faecalis*, *Pseudomonas aeruginosa*, *Staphylococcus aureus*, *Proteus mirabilis*, and *Klebsiella pneumoniae* [29,30]. Pathogenic fungi like *Candida* spp. are also reported in these infections. Pathogens involved in CAUTIs can originate from two different sources, either endogenous bacteria from gastrointestinal and vaginal flora or exogenous bacteria, which can enter through urinary catheterization [30]. Therefore, aseptic insertion techniques and good management are indispensable to reducing the risk of introducing uropathogens into the urinary tract [31]. The ability of biofilm-embedded bacteria to resist and tolerate antimicrobials significantly contributes to the development and persistence of chronic bacterial infections, particularly those associated with UCs [32]. The majority of antimicrobial therapies that are now on the market are typically developed and evaluated against microorganisms that live in the form of planktonic cells [33]. As a result, these therapies frequently fail to eradicate biofilm-related infections [34,35]. Moreover, in CAUTIs, the interplay between host immune responses and bacterial biofilms plays a critical role in infection persistence [36]. Biofilms that form on catheter surfaces shield bacteria from immune clearance by encasing them in an extracellular polymeric substance (EPS) matrix that resists phagocytosis and complement activity [37], and immune cells are unable to fully penetrate the biofilm matrix, leading to incomplete eradication of pathogens [38]. Therefore, effective anti-biofilm therapeutic strategies must be developed to combat CAUTIs, which frequently exhibit resistance to conventional therapies. These novel non-antibiotic strategies include bacteriophages and their derived enzymes that selectively lyse biofilm-forming bacteria, as well as antimicrobial peptides that disrupt bacterial membranes and modulate quorum sensing. In addition, phytochemicals with anti-biofilm properties and surface modifications of catheters, such as anti-adhesive, galvanic, or nanoparticle coatings, are designed to inhibit bacterial adhesion and prevent early colonization [39,40,41,42,43,44].

Developing successful treatments to target biofilm infections represents a significant and urgent challenge in the field of antibacterial drug research. Therefore, understanding the relationship between bacterial biofilms and antimicrobial resistance is crucial to effectively treating biofilm-associated infections, including CAUTIs. This review aims to provide an update on CAUTIs, discuss the role of biofilms in sub-optimal responses to antimicrobial treatment, and present some new experimental and clinical approaches to better understanding and combating these serious infections.

## 2. Bacterial Biofilm Formation and Encrustation

A bacterial biofilm is an organized community of bacterial cells that firmly adhere to abiotic or biotic surfaces and are encased in a matrix made of extracellular polymeric substances (EPSs) [45]. More importantly, abiotic surfaces, including UCs, are more prone to colonization by bacteria and build-up of biofilm compared with biotic surfaces. This susceptibility comes from the significantly lower bacterial inoculum required for colonization of abiotic surfaces than host tissues (~10,000-fold less) [46]. Another reason is that implants are not vascularized structures compared with normal tissues, making them more vulnerable to colonization [47]. Even though biofilm formation on different surfaces is a complex and dynamic process, different authors often identify four main stages of bacterial biofilm formation, including bacterial attachment (first reversible and then irreversible attachment), microcolony formation, maturation, and dispersal of the biofilm with subsequent colonization of new environments, as shown in Figure 1A. The surrounding EPSs function as a multi-layered defense system for the biofilm community [48]. Additionally, the EPSs provide structural support and resist the physical forces exerted by the surrounding environment (e.g., flowing liquid), ensuring the biofilm’s stability [49]. The majority of human infections, particularly chronic and recurring ones, stem from bacteria capable of forming biofilms [48]. The physical and functional features make biofilms formed on implants very difficult to treat, and several authors compare them to shielded fortresses [50].

Biofilm formation on UCs is influenced by the specific characteristics and topology of the catheter, which contribute to the adherence and proliferation of bacteria on its surface [51]. The catheter’s material and surface roughness provide an ideal substrate for microbial adherence and colonization [52]. For example, the most common uropathogen, *E. coli*, demonstrates a greater affinity for rough surfaces. Extensive research indicates that this bacterium exhibits notably higher adherence to roughened polystyrene in comparison to its smooth counterpart. This heightened adherence facilitates stronger initial attachment for *E. coli*, potentially paving the way for biofilm formation and colonization [53]. Additionally, the intricate design of catheters creates gaps and microenvironments that facilitate the attachment of microorganisms, promoting the establishment of biofilm communities [54]. As urine flows through the catheter, it introduces a constant supply of nutrients that sustains microbial growth, further contributing to biofilm development [55]. The formation of conditioning layers, comprising host proteins, including fibrinogen, fibronectin, collagen, and microbial by-products, adds a layer of complexity to the biofilm matrix, enhancing its resilience and resistance to antimicrobial agents [56]. Additionally, latex catheters may feature embedded diatom skeletons, serving as sites for bacterial attachment [54]. Catheter-associated biofilms may be formed by a single species or a mixture of bacteria, but prolonged catheterization promotes polymicrobial communities [10]. Fewer studies have been performed on polymicrobial CAUTIs and biofilm development since modeling mixed-species biofilms is more challenging than modeling those created by a single species; therefore, further studies are needed in this area [57].

Encrustation represents a significant issue that contributes to numerous chronic and recurrent complications associated with the prolonged placement of urinary catheters [57]. Within the urinary system, crystal deposits can result from a variety of physiological processes and have a diverse composition. They can be classified into five types based on the major component: uric acid, calcium oxalate, calcium phosphate (carbonate-apatite), cystine, and magnesium ammonium phosphate (struvite). The first four kinds are called metabolic encrustations because they result from dysfunctions of metabolic processes, whereas magnesium ammonium phosphate (struvite) encrustation is caused by microbial infections [58]. Chemical interactions take place between the negatively charged biofilm matrix and positively charged ions like magnesium and calcium. These interactions aid in the crystallization process of magnesium and calcium phosphates, ultimately resulting in the formation of crystalline deposits [59]. Variations in the levels of electrolytes and the composition of urine can significantly influence the formation of biofilms [60].

Microbial crystalline deposits form through a series of stages. Initially, the urinary tract is infected by urease-positive bacteria like *P. mirabilis* or *K. pneumoniae* (to a lesser degree, *P. aeruginosa*) [61,62]. These bacteria subsequently adhere to the catheter surface, initiating biofilm formation, as depicted in Figure 1B. Urease breaks down urea into ammonia and carbon dioxide, which contributes to an increase in urine pH, facilitating the formation of hard crystals, which become integrated into the biofilm matrix alongside bacteria. Driving factors such as the catheter’s surface characteristics, mineral deposit expansion, and continuous nutrient flow from urine contribute to this build-up of biofilm on the urinary catheter. Accumulation of these deposits within the catheter lumen may obstruct urine flow, resulting in complications like bladder distension, catheter leakage, and urine reflux to the kidneys, potentially leading to ascending infections [63].

## 3. Biofilm Matrix Structure and Its Composition

The architecture of the biofilm matrix is influenced by different factors, such as bacterial species participating in biofilm formation, the composition of the surrounding growth medium [64], and host factors such as fibrinogen deposition on catheters as a result of catheter-induced bladder wall irritation [65]. Recent studies on mouse and human urinary catheterization have revealed the crucial role of host clotting factor 1, fibrinogen (Fg), in surface adhesion. This process contributes to the formation of biofilms and the persistence of CAUTIs caused by *E. faecalis* and *S. aureus*. Fg is continuously released into the bladder lumen as a response to the mechanical injury to the urothelial lining caused by catheterization. Upon entering the lumen, Fg is deposited on the catheter surfaces, forming a structure that facilitates the attachment of uropathogens and the onset of CAUTIs in both humans and mice [65,66,67,68]. Moreover, the availability and diversity of nutrients within the growth medium further shape the biofilm structure, affecting factors like thickness, porosity, and overall stability. In in vitro *P. aeruginosa* biofilm models, a common feature is the development of mushroom-like structures. These structures, along with surrounding empty spaces, emerge as the biofilm matures and are influenced by environmental factors such as nutrient availability and iron levels [69]. In contrast, flat biofilm morphologies are often seen in other species, like *S. aureus*. The biofilm matrix is believed to have a sponge-like structure, where water is held in pores and channels [70]. The flow and viscosity of this fluid are crucial to transporting nutrients and waste, supporting the biofilm’s biological functions [71].

Biofilm architecture investigations have elucidated the presence of microbial cells and extracellular polymeric substances (EPSs), which constitute up to 90% of the dry biomass and are considered the primary constituents, while bacterial cells make up the remaining portion. Although EPSs differ in their physicochemical properties, they are mainly composed of polysaccharides [72]. A significant portion of the total biofilm matrix, up to approximately 97%, is composed of water, although the exact composition may vary depending on the specific system being studied [73]. While the physical and chemical makeup of the EPSs may differ among bacterial species, it primarily consists of polysaccharides, proteins, lipids, and extracellular DNA (eDNA) [74], as summarized in Table 1. These components intertwine, forming a dynamic meshwork that encases the cells, creating a three-dimensional microenvironment [35]. Furthermore, EPS production and organization differ between single-species and multi-species microbial communities. Environmental factors like nutrient availability, the presence of antimicrobials, and flow conditions can influence the composition of the biofilm matrix [75].

Polysaccharides make up a significant portion of the EPS matrix in biofilms and are typically heteropolysaccharides composed of both neutral and charged sugars, often incorporating organic and inorganic elements that influence their charge. The composition of EPSs varies among species and even strains [74]. In UTI-related biofilms, key polysaccharides include cellulose, alginate, poly-β-,1,6-N-acetylglucosamine (PNAG), and capsular polysaccharides [74,76]. Extracellular proteins are also key components of the biofilm matrix and can sometimes be more abundant than polysaccharides. Structural proteins support the stability of the biofilm and aid in cell-to-matrix interactions, while enzymes degrade components like polysaccharides, proteins, and eDNA to provide nutrients and enable biofilm remodeling. For example, in *E. faecalis* biofilms, proteins such as aggregation substance (Agg), gelatinase (GelE), and enterococcal surface protein (Esp) are vital to bacterial aggregation, surface adhesion, and biofilm formation [77]. Additionally, extracellular DNA (eDNA) plays a critical role in maintaining the integrity and antimicrobial resistance of *S. aureus* biofilms, with its disruption leading to reduced biofilm formation and increased sensitivity to treatments [78].

## 4. Quorum Sensing Regulation of Biofilm Formation in Bacteria

Within the bacterial community, numerous cellular processes, such as population density-dependent virulence factors, genetic material transfer between cells, biofilm formation, drug resistance, and secondary metabolite synthesis, are regulated by a chemical communication system called quorum sensing (QS) [79]. QS is a method through which bacteria communicate with each other, employing signaling molecules known as autoinducers (AIs) that they produce and release into external environments. As the bacterial population increases, the concentration of these signaling molecules also increases. Once a certain level of these AIs is reached, bacteria coordinate their activities, affecting behavior like biofilm formation and virulence [80]. While Gram-positive and Gram-negative bacteria possess QS mechanisms, the specific signaling molecules they utilize for communications vary between them [81].

During biofilm formation, the QS system serves as a crucial regulatory mechanism to control the build-up of biofilm. In Gram-negative bacteria, the QS system employs acyl homoserine lactone (AHL) as a signaling molecule to regulate biofilm formation, consisting of specific signaling molecules and their corresponding receptors. For instance, in *P. aeruginosa*, the QS system comprises two distinct signaling pathways: lasI/lasR and rhlI/rhlR. The genes lasI/rhlI encode synthetases for different signaling molecules, while lasR and rhlR encode receptors for these respective molecules [82,83]. As bacterial density increases, the secretion of signaling molecules also rises. Upon reaching a specific threshold, these signaling molecules bind to their corresponding receptors, activating them. Activated receptors then stimulate relevant transcriptional regulators to produce extracellular polysaccharides, toxins, alginate, and other factors, prompting bacteria to initiate biofilm formation [84,85].

In uropathogenic *E. coli* (UPEC), the quorum sensing system plays a critical role in biofilm formation during UTIs. This system involves the synthesis of the signaling molecule AI-2 by the enzyme LuxS, which facilitates communication between *E. coli* cells and potentially with other bacterial species. This intercellular communication is essential to coordinating the expression of genes that contribute to biofilm development, enabling *E. coli* to establish persistent biofilms on the urinary tract lining. The LuxS/AI-2 pathway is essential to the growth and maturation of *E. coli* biofilms [86,87]. Similarly, the QS system of *E. faecalis* is known to play a crucial role in the formation of biofilm and the production of virulence factors. *E. faecalis* is also implicated in polymicrobial infections, and it can facilitate biofilm formation by other Gram-negative bacteria. For example, *E. faecalis* enhances *E. coli* biofilm formation under iron-limited conditions, promoting polymicrobial growth [88]. Among the key regulators of *E. faecalis* QS, gelatinase (GelE) and serine protease (SprE) have been identified as significant contributors. GelE and SprE are recognized as virulence factors that facilitate the degradation of host proteinaceous substrates, promote biofilm formation, and enhance adhesion [89,90,91].

In Gram-positive bacteria, the QS system controls biofilm formation by using oligopeptides as signaling molecules. These molecules undergo modification and are recognized by two-component sensing proteins, which, through phosphorylation and dephosphorylation, regulate the expression of target genes, thus influencing biofilm formation. Different bacteria utilize distinct oligopeptide signaling molecules and QS system regulation pathways [79]. The regulatory system in *S. aureus* has a two-component structure, where a sensor protein detects the environment and activates a regulatory protein. For example, the WalK/WalR system in Gram-positive bacteria directly activates biofilm formation genes [92]. In addition to two-component systems, there are also known regulatory factors that influence biofilm formation. In *S. aureus*, a specific type of RNA polymerase helps build biofilms, while another type hinders them [93,94]. Interestingly, some phytochemicals primarily hinder the quorum sensing process by obstructing quorum sensing inducers like AHL and autoinducer type-2 [95]. Many researchers have argued that quorum quenchers, in conjunction with antibiotics, represent the most promising therapeutic strategy against biofilm formation [96].

## 5. Antimicrobial-Induced Biofilm Formation

Bacteria may be exposed to sub-minimal inhibitory concentrations (sub-MICs) of antimicrobial agents at various points during a treatment regimen. This may occur at the start and the end of dosing, between doses, or during low-dose therapies and has significant clinical implications in UTIs, among others [97]. Antimicrobial agents effectively hinder pathogen growth when their concentration surpasses the MIC between doses [98]. Aminoglycosides display concentration-dependent bactericidal activity, with previous studies reporting that achieving a high maximal concentration (Cmax)/MIC ratio leads to greater clinical efficacy. In more recent data, the area under the concentration–time curve (AUC0–24)/MIC ratio may be a superior predictor of activity with extended-interval dosing [99]. In contrast, the effectiveness of time-dependent antimicrobials (e.g., beta-lactams) depends on the duration that the drug concentration remains above the MIC (T>MIC) during a dosing interval. The longer the drug concentration stays above the MIC, the more effective the treatment [100,101]. The concept of the abovementioned pharmacokinetic/pharmacodynamic (PK/PD) indices has been developed based on studies on planktonic bacteria, but whether it may be fully applied to biofilms is not yet entirely clear [102].

PK/PD indices play a vital role in developing effective antimicrobial dosing strategies. Since drug concentrations decrease over time as the body eliminates them, concentrations may decrease below therapeutic thresholds between doses. When this occurs, susceptible bacteria may continue to multiply despite the drug being present, ultimately compromising the success of the treatment.

Studies have shown that exposing bacteria to sub-MICs of antimicrobial agents can trigger various effects in microorganisms [103]. These effects include metabolic stress, increasing bacterial resistance to higher antibiotic doses, promoting the production of enzymes and toxins, inducing genetic and ultrastructural changes, enhancing adhesion properties and virulence factors, and modulating the expression levels of genes associated with biofilm formation [98,104]. This phenomenon holds clinical significance in many types of infections, including UTIs. Although many antimicrobials used in UTIs achieve high urinary concentrations, with lower doses or less frequent administration, bacteria may be exposed to sub-MIC levels of these drugs [105]. For example, a sub-MIC of tobramycin was found to trigger the formation of *P. aeruginosa* biofilm, likely through a mechanism involving the intracellular signaling molecule cyclic dimeric guanosine monophosphate [106]. Similarly, *P. aeruginosa* exposed to a sub-MIC of antibiotics such as ciprofloxacin or tobramycin has been shown to increase biofilm formation [107]. *E. coli* can display biofilm induction over a broad range of tetracycline concentrations, typically ranging from 0.1 to 0.8 times the MIC [108]. Consequently, infections involving biofilm-forming bacteria can be more persistent, recurrent, and challenging to treat. Of particular concern is the antimicrobial exposure of preformed biofilms for which the MIC level typically does not represent the susceptibility threshold. Since biofilm-embedded bacteria are often dormant and protected by the matrix, it is expected that the adaptive mechanisms may be triggered at much higher drug concentrations that would be lethal for planktonic bacteria. Understanding this phenomenon and choosing appropriate antimicrobials is crucial to developing effective strategies (e.g., dosage protocol optimization) to combat biofilm-associated infections in clinical settings.

## 6. Mechanisms of Antimicrobial Resistance in Biofilm-Forming Bacteria

Globally, multidrug-resistant bacteria are thought to be responsible for 700,000 human fatalities annually; by 2050, this number is predicted to surpass 10 million deaths, at a total cost of USD 100 trillion [78]. The formation of biofilms on medical devices like urinary catheters poses a significant challenge, largely due to bacterial persistence (lack of sensitivity to antimicrobials due to dormant state) [109] and phenotypic tolerance to antimicrobials [110]. Both these phenomena may contribute to the development of antimicrobial resistance. Antimicrobial resistance and biofilm formation are important causes of ineffective antimicrobial therapy; however, the mechanisms involved differ significantly. Resistance can develop through genetic mutations, acquisition of resistance genes from other organisms, or selective pressure exerted by the use of antimicrobial agents [110]. However, biofilms on urinary catheters act as miniature ecosystems, shielding bacteria from antimicrobials through complex physical barriers and physiological adaptations [111,112]. Despite extensive research on drug delivery through urinary catheter-related biofilms, our understanding of the exact mechanisms behind treatment failure is still limited. Therefore, comprehending the strategies employed by bacteria embedded in biofilm to withstand antibiotics would facilitate the development of effective surveillance protocols and novel strategies to tackle biofilm-related infections [111]. Here, we review the main mechanisms of antimicrobial resistance in biofilms.

### 6.1. Reduced Penetration

Biofilm treatment effectiveness relies on antimicrobial agents’ ability to penetrate the biofilm’s diverse structure on the urinary catheter, which is hindered by its mechanical and physicochemical properties [112,113]. The exterior layers of the biofilm serve as a protective barrier, absorbing and redirecting antimicrobial substances away from the deeper layers where bacterial cells are situated [114]. This stratified arrangement establishes a gradient of diffusion, resulting in a progressive reduction in the antimicrobial effect as they penetrate deeper into the biofilm. For instance, this phenomenon was observed for aminoglycosides, which are hydrophilic and positively charged antibiotics [115]. The interaction between the negatively charged matrix and positively charged aminoglycosides prevents these antibiotics from entering the deeper layers of bacterial biofilms and limits their efficacy [116]. Similarly, it has been discovered that eDNA, a recognized component of biofilms, may contribute to reduce antibiotic penetration to the deeper region of the biofilm [117,118]. Negatively charged eDNA can capture positively charged molecules, including some antibiotics. This trapping effect reduces the amount of antibiotics that can reach and kill the bacteria embedded in the biofilm matrix. When exposed to antimicrobials, bacteria at the biofilm surface are likely to die and contribute to the increase in EPS thickness; however, bacteria embedded deeper within the biofilm are shielded by the matrix and will likely survive. The penetration of antimicrobials into the biofilm may also be restricted by physicochemical features like molecular weight, charge, and hydrophobicity [119].

### 6.2. Physiological Heterogeneity and Reduced Metabolic Activity

Bacterial biofilms exhibit distinct physiological heterogeneity and reduced metabolic activity compared with planktonic bacteria, which contribute to the formation of persister cells [120]. Within the biofilm, bacteria adjust to specific microenvironments with variable oxygen availability, nutritional gradients, and other environmental elements, resulting in increased physiological heterogeneity. These conditions contribute synergistically to antimicrobial resistance through various mechanisms [96]. For instance, a biofilm formed by *E. coli* in the urinary tract may contain dormant or slow-growing cells, making them less susceptible to antibiotics like fluoroquinolones, which primarily target dividing and metabolically active bacteria [121].

Moreover, the reduced growth rate and altered metabolic functions of bacterial biofilm cells play a crucial role in their increased antibiotic resistance [122,123,124]. The biofilm matrix creates a complex environment where nutrient and oxygen levels vary significantly, resulting in unique metabolic conditions for each cell, enabling specific subpopulations of bacteria to withstand antimicrobial treatments [125]. Cells in the middle of the biofilm get less oxygen, so they grow slowly and have low metabolic activity [126]. This makes them more resistant to antibiotics, which target active cells. Unlike genetically resistant cells, persister cells exhibit a temporary form of resistance that stems not from genetic mutations but from their metabolic inactivity and dormancy, enabling them to evade the effects of antibiotics [127]. Persister cells’ tolerance is a reversible state, which enables them to regain metabolic activity and resume growth once antibiotics are removed [128].

Research indicates that bacteria within biofilms exhibit a growth rate approximately one-third that of planktonic cells during the exponential growth phase [129]. The reduced growth rate observed in *P. aeruginosa* biofilms aligns with findings across multiple bacterial species, highlighting the slower replication of cells associated with biofilms [129]. This diminished growth, particularly within the deeper layers of the biofilm, compromises the efficacy of numerous antibiotics, which are typically designed to target actively dividing cells [130]. Genomic investigations have elucidated the relationship between reduced bacterial growth rates and antibiotic resistance within biofilm-associated populations. For instance, in *P. mirabilis*, the downregulation of genes such as *pst*S, *sod*B, and *fum*C has been shown to correlate with slower growth and decreased metabolic activity. These genetic and metabolic shifts contribute to enhanced antibiotic resistance, highlighting the remarkable resilience of biofilm communities and their ability to escape therapeutic strategies [131]. Consequently, a comprehensive understanding of these adaptive mechanisms is critical to designing effective intervention approaches.

### 6.3. Efflux Pumps and Enzymatic Modifications

Biofilms employ multiple mechanisms to render drugs ineffective, including efflux pumps and enzymatic modifications [132], and these specialized mechanisms act as defense systems against antibiotic action. Efflux pumps play a complex role in biofilm formation; some efflux pumps help bacteria adhere to surfaces and produce the components of the biofilm extracellular matrix, thus promoting its formation and growth. Conversely, some efflux pumps can hinder biofilm formation by interfering with chemical signals that bacteria use to communicate and regulate biofilm genes [133]. Exposure to non-lethal levels of antimicrobial agents has been shown to trigger the activation of multidrug efflux pumps and lead to the emergence of efflux pump mutants [134].

Although they may be responsible for the antimicrobial-resistant phenotype in planktonic cells, efflux pumps may also actively remove antibiotics from the bacterial interior within the biofilm [133]. Efflux pumps in the biofilm environment, which may be expressed at higher levels than in planktonic cells 134,play a key role in enhancing bacterial survival by actively extruding antimicrobial compounds [135]. This reduces the effective concentration of the drug [136]. For instance, *P. aeruginosa* possesses multiple efflux pump systems, including MexAB-OprM, MexXY-OprM, and MexCD-OprJ, which effectively eliminate antibiotics including most β-lactams from the cell [137]. In the context of biofilm formation, the expression of the MexAB-OprM, MexXY-OprM, and MexCD-OprJ genes has been found to be upregulated, which is believed to contribute to the overall resistance of the biofilm-embedded bacteria [138]. Similarly, *S. aureus* also utilizes efflux pumps like NorA and NorB to expel a broad spectrum of antibiotics, rendering it resistant to beta-lactams, fluoroquinolones, and other antibiotic classes [139]. Studies have shown that the expression of the norA and norB genes is upregulated during biofilm formation in *S. aureus*. The increased expression of these efflux pumps in biofilm-associated cells suggests their involvement in the adaptive response to antimicrobial stress within the biofilm environment. The upregulation of the norA and norB genes in biofilms may confer increased resistance to antibiotics commonly used in UTIs [140].

Similarly, the biofilm matrix can contain enzymes that degrade antimicrobial agents, as observed with β-lactamases produced by bacteria such as *P. aeruginosa*, *K. pneumoniae*, and *E. coli*. These enzymes are capable of breaking down penicillins, carbapenems, and other β-lactam antibiotics, thereby contributing to antimicrobial resistance [141]. For example, *K. pneumoniae* biofilms secrete β-lactamase, which has been shown to effectively degrade ampicillin and prevent it from penetrating the biofilm and reaching the embedded cells [142]. In *P. aeruginosa*, a chromosomally encoded AmpC β-lactamase is secreted into the biofilm matrix and plays a significant and clinically relevant role in β-lactam resistance. A study conducted by Hall, C.W. and Mah, T.F. (2017) revealed that exposure to imipenem and piperacillin triggers β-lactamase production in *P. aeruginosa* biofilms [143].

### 6.4. Acquisition and Transmission of Antimicrobial Resistance Genes in Biofilms

Biofilms are recognized as a key factor in the dissemination of antibiotic resistance genes (ARGs) due to the proximity of bacteria within the biofilm, which increases cell-to-cell contact and, consequently, the likelihood of genetic exchange [144]. In biofilms, susceptible bacteria can protect themselves against antimicrobials through two strategies: spontaneous mutations in their genetic code or acquisition of resistance genes from others through horizontal gene transfer (HGT) [145]. HGT is considered a larger threat due to its rapid and efficient spread within the dense biofilm community which allows many bacteria to become resistant much faster than individual mutations would permit [146,147]. Numerous investigations have revealed that biofilms may serve as hotspots for the spread of ARGs [148]. Therefore, biofilms harbor and share ARGs, acting as breeding grounds for “superbugs” resistant to antimicrobial agents. Also, mobile genetic elements (MGEs) are specific DNA segments enabling their transfer within the host genome (intracellular mobility) or among bacterial cells (intercellular mobility) [149]. The exchange of DNA fragments between a cell donor and a receptor occurs through processes such as conjugation, transformation, or transduction [150], as summarized in Table 2.

## 7. Model Systems to Study Biofilms Under In Vitro and In Vivo Conditions

Over recent decades, there has been increasing attention towards constructing models to enhance the understanding of the mechanisms of biofilm formation and optimize therapeutic strategies against biofilm-related infections, including CAUTIs [156,157]. These models are categorized into in vivo and in vitro model systems. In vivo models have the potential to reflect the physiological conditions observed in humans and animals, making them highly relevant for translational research [158,159]. They also help to study the interaction between the host immune system and biofilm infections [160]. On the other hand, in vitro models are essential to complementing in vivo studies and clinical trials, offering more accurate insights into the efficacy of new treatments [161]. They are generally less labor-intensive, their higher throughput minimizes laboratory animals’ suffering (compliance with 3R principles), and they require less specialized equipment and are cheaper.

### 7.1. In Vitro Biofilm Models

In vitro biofilm models are simplified versions of biofilms grown entirely in the laboratory, often mimicking specific aspects of natural biofilm on different surfaces [162]. To reflect the clinical conditions observed in UTIs, it is essential to capture the characteristic parameters that describe in vitro models. These parameters include the physiologically relevant medium (pooled human or artificial urine), hydrodynamic conditions (urine flow rate and shear stress), the growth substrate reflecting catheter materials, and the typical bacterial species causing UTIs [159,163]. These models are valuable because they enable researchers to easily study basic principles of biofilm formation, physiology, and structure. There are many different types of in vitro biofilm models available, ranging from microtiter plates to complex systems that take into account the physiological characteristics needed to represent biofilm development in specific environments [164]. Depending on the availability of nutrients over time and elimination of waste from the systems, the range of in vitro models may be divided into two primary groups: static (closed) and dynamic (open) models [164], as shown in Figure 2 below. They are classified based on whether the drug concentration stays constant or changes over time [165]. Exposure of bacteria to a constant antimicrobial concentration in an in vitro model does not account sufficiently for clinical scenarios [166]. In contrast to this, within in vitro PK/PD flow models, the concentration of the antibiotic over time changes, resulting in bacteria being exposed to fluctuations resembling real-life PK profiles seen in patients [167,168,169].

#### 7.1.1. Static or Closed Models

These are the most common and simplest types of biofilm models. Static models have limited nutrient supply since the medium remains unchanged throughout the biofilm growth phase in a microtiter plate and other static model systems. They consist in growing bacteria in a well or on a specific surface with limited nutrients and aeration. This mimics environments where biofilms naturally form, such as on indwelling medical devices, including UCs [170]. Common static models include microtiter plates (12-, 24-, 48-, and 96-well plates), Petri dishes, the Calgary Biofilm Device (CBD) (now known as the MBEC Assay^®^ Kit) (Innovotech Inc., Edmonton, Canada), and the Biofilm Ring Test (BRT) [171]. These techniques are simple, rapid, and inexpensive and enable fast assessment of biofilm mass using staining methods such as crystal violet or the quantification of viable cells through assays like reduction in tetrazolium salts, e.g., XTT (2,3-bis-(2-methoxy-4-nitro-5-sulfophenyl)-2H-tetrazolium-5-carboxanilide) [171]. However, since, in these systems, bacterial cultures sediment and are not subjected to the shear pressures that are common in many in vivo locations (e.g., urinary system), they are often not regarded as real, mature biofilms. Furthermore, because the culture medium is often only manually changed every 12 to 48 h depending on the purpose of the experiments, static cultures typically suffer from limited availability of nutrients, and the accumulation of bioproducts may result in toxicity [172].

#### 7.1.2. Dynamic (Open) Systems

These models attempt to replicate the shear stress and nutrient flow conditions that biofilms experience in their natural environment [173]. Flow models can be continuous or intermittent, and they can be used to study the effects of different flow rates and shear stresses on biofilm formation and detachment [174]. Dynamic models are more complex than static models, but they can provide more realistic data on biofilm behavior, as they mimic the real host conditions [175]. These models support biofilm growth with a steady supply of nutrients and are ideal for studying biofilms on urinary catheters because the continuous flow replicates the urine that coats the catheter surface. The most commonly used dynamic models include Hollow Fiber Infection Model (HFIM), drip flow biofilm reactor (DFBR), Modified Robbins Device (MRD), Rotating Disk Biofilm Reactor (RDR), and microfluidic systems [176]. In these systems, biofilms build-up under specific hydrodynamic conditions on intraluminal surfaces like UCs or flat surfaces like silicone coupons or microscopic glass slides, which can be extracted for further analysis [177].

More importantly, in vitro dynamic models are excellent for studying the PK/PD indices related to UTIs and for guiding optimized dosing schedules, clinical trials, and treatment guidelines [159,169]. This is typically achieved by setting up a system of containers where one contains a high concentration of a given drug, whereas the other one, filled with clear fresh medium, dilutes the drug over time. By using precise programmable peristaltic pumps, one can easily mimic first-order PK process with virtually no elimination rate constant, which enables the simulation of different dosage protocols and resulting fluctuations in drug concentrations that are observed in treated UTI patients (input data can be obtained from published studies involving human subjects). These systems can also account for the hydrodynamic conditions (urine flow) present in CAUTIs [80,165,178] and urinary bladder voiding [179]. Although the majority of dynamic models focus on bacteria in the planktonic phase, a few of these models were created specifically to investigate bacterial biofilms [180]. Therefore, the development of a dynamic CAUTI in vitro PK/PD model to simulate the PK profiles of common drugs used in the treatment of biofilm-related infections is needed. A simple dynamic in vitro PK/PD flow model applying a drip flow biofilm reactor designed for investigating antimicrobial agents’ effects on biofilms is indicated in Figure 3 below.

Moreover, dynamic in vitro models, while valuable for simulating physiological medium flow dynamics, have more limitations compared with models specifically designed for dynamic bladder infection in vitro modeling [181]. A key shortcoming is their inability to replicate the specific microenvironment of the urinary bladder [182]. Furthermore, many of these models rely on flat coupons or simplified flow chambers, which inadequately mimic the complex urinary flow dynamics and biofilm development on indwelling catheters, critical features of CAUTIs [182,183]. Instead of reproducing the intermittent urine flow of the urinary tract, they typically employ continuous flow, thereby reducing physiological relevance [184,185,186]. In contrast, specialized dynamic bladder infection in vitro models incorporate intermittent flow and other urodynamic parameters, providing a more representative platform for UTIs [182,183,186,187].

Although recent advances in dynamic in vitro systems have improved experimental control and reproducibility, they still fail to capture several key aspects of UTIs related to host factors [182]. The formation, maturation, and dissemination of biofilms arise from complex interactions between bacterial communities and their host environments in vivo, which are difficult to reproduce under in vitro conditions [187]. These models cannot fully replicate the host immune response, mucosal architecture, or urinary composition that shape bacterial behavior in patients, often leading to discrepancies between in vitro findings and clinical outcomes [181]. The complex interplay of host factors, catheters, and bacterial communities also remains difficult to model accurately [187]. While they can reproduce flow conditions and nutrient gradients, these systems do not account for polymicrobial interactions or host-derived proteins, limiting their predictive validity and necessitating confirmation in in vivo or clinical settings [188]. Similarly, PK/PD infection models replicate human drug concentration–time profiles but neglect host clearance mechanisms, immune contributions, and tissue-level drug distribution. Collectively, these limitations reduce the predictive accuracy of dynamic in vitro models for therapeutic outcomes [189] and constrain the direct translation of findings into clinical practice without intermediate validation in in vivo studies [190].

#### 7.1.3. Dynamic Bladder Infection In Vitro Models

A dynamic bladder infection in vitro model is a laboratory system designed to simulate the physiological conditions of the bladder and study urinary tract infections (UTIs), including planktonic uropathogens in simulations of uncomplicated UTIs and biofilm-forming uropathogens in CAUTIs [159,191]. Moreover, these models have been developed to study the efficacy of antimicrobial agents, understand bacterial biofilm formation, and optimize treatment strategies [169]. These models facilitate controlled studies of bacterial colonization, the comprehension of the basic principles regulating microbial biofilm formation, and therapeutic interventions within a simulated bladder environment [159,192]. Unlike static models, they mimic physiological conditions by incorporating fluid flow, pressure changes, bladder voiding, and urine turnover, closely resembling the bladder environment. Key components of these models typically include peristaltic pump systems programmed to mimic urodynamics, artificial urine media, and biomaterials designed to mimic the physiochemical properties of the bladder. More advanced systems may even include cultured uroepithelial cells as a natural interface for the host–bacteria interactions [193]. Bladder model systems have demonstrated that microbial biofilm formation is significantly influenced by flow dynamics, nutrient accessibility, nutritional composition, and other physicochemical features, accurately reflecting the physical and chemical parameters present in a catheterized human bladder [194,195]. Furthermore, while credibly simulating many clinical conditions, these advanced in vitro models bypass the ethical issues, complexity, and high expenses of in vivo research. However, in vitro models also have their limitations. They cannot capture all the dynamics and complexity of the interactions between the host, especially the involvement of the immune system, and the biofilm-forming bacteria, which represents the major drawback of these systems [159].

### 7.2. In Vivo CAUTI Bacterial Biofilm Models

Due to the aforementioned limitations of in vitro setups, establishing reliable in vivo models is crucial to confirming findings from in vitro studies and is an essential stage in evaluating new treatments and devices for potential clinical use [196]. Decades ago, it was noted that systemic antibiotics were ineffective in eliminating bacteriuria unless the urinary tract device was removed [197]. In 1985, an electron microscopy examination of a urethral catheter, which was removed due to recurring CAUTIs, unveiled the existence of a bacterial biofilm [198]. Since then, numerous in vivo models have been devised to replicate these scenarios. Many of these models involve introducing a foreign object into the bladder, with or without surgical intervention [171]. Surgical models rely on the surgical insertion of a foreign body inside an animal bladder such as glass beads or pieces of UCs [199]. The underlying concept of non-surgical models involves the transurethral insertion of catheter fragments into the animal’s bladder. Such models have been established in laboratory animals to investigate different aspects of CAUTIs, including bladder inflammation, the virulence of *E. faecalis* on silicone implants, and preventive strategies like Triclosan^®^ (Kumar Products Ltd., Bangalore, India) (a broad-spectrum antiseptic) [200].

In humans and mice, prolonged catheter uses causes ongoing bladder inflammation, leading to the build-up of fibrin and fibrinogen on the catheter over time. This accumulation damages the urothelium and creates a surface that facilitates the formation of biofilms by pathogens responsible for CAUTIs [65,67,201]. The study by Flores-Mireles and colleagues demonstrates that endocarditis and biofilm-associated pilus (Ebp) functions as an adhesin, enabling bacteria to adhere to host fibrinogen. This fibrinogen is released and deposited onto catheters following their insertion into the mouse bladder [65]. Various therapeutic interventions have been evaluated in these models, such as antibiotics alone or in combination with small molecules (such as mannosides) to prevent CAUTIs. However, the effectiveness of this adjunctive approach still requires validation in clinical settings [202].

However, in vivo model systems are associated with some limitations. One of the significant drawbacks is associated with ethical concerns related to the use of laboratory animals for experimentation. The welfare and rights of animals must be carefully considered, and alternative methods should be explored whenever possible. Moreover, in vivo experiments are often time-consuming, expensive, and often characterized by lower throughput and require specialized facilities and expertise. These models frequently employ monoculture biofilms, a scenario rarely encountered in natural settings [203,204]. In addition to the previously mentioned limitations, the PK aspect of this model generally cannot accurately replicate human PK profiles and is usually confined to the natural PK characteristics of the animal species used [205].

Conventional in vivo models are critical to studying infections by providing a living system to observe how pathogens establish infections and form specialized structures such as intracellular bacterial communities and quiescent intracellular reservoirs and to investigate the role of the immune response in shaping infection dynamics [206,207]. These traditional models, often using female mice of genetically uniform inbred strains, have provided valuable insights into bladder infection mechanisms. However, their translational relevance is limited by anatomical and immunological differences from humans, such as variations in urothelial structure, urine composition, and immune response [208]. To address these limitations, researchers have increasingly turned to genetically modified and humanized models. For example, mice deficient in Atg16L1 (Atg16L1^HM2^) lack a critical autophagy-related gene, and upon transurethral inoculation with UPEC, they exhibit altered infection dynamics [209]. This highlights the contribution of host autophagy to bacterial persistence and immune control within the bladder [210]. Similarly, RNASE3 transgenic mice, which incorporate a human immune component, demonstrate enhanced resistance to ascending *E. coli* infections, underscoring the value of integrating human-specific factors in preclinical studies [211].

Advances in humanized models, particularly those featuring functional human immune systems or engineered bladder tissue, are providing more accurate platforms to study UTI pathogenesis and therapeutic responses [212]. These models enhance preclinical predictability, bridging the gap between conventional animal studies and clinical trial outcomes [6,212]. Collectively, such innovations have significantly improved the translational relevance of UTI research, enabled deeper mechanistic insights and supported the development of novel antimicrobial strategies [212].

## 8. Novel Strategies to Combat Bacterial Biofilms

Different clinical practice guidelines highlight the need for effective management approaches for patients with indwelling UCs to reduce the risk of biofilm development and CAUTIs [213]. These preventive measures are indispensable in clinical practice. However, therapeutic strategies are required to treat established biofilm-related infections. Conventional antibiotic therapies frequently fail to eliminate biofilm-related infections [214]. Because biofilm-related infections involve a significantly different form of bacterial growth compared with planktonic cells, the development of dedicated and innovative methods is required based on our understanding of the intricate nature of biofilms. Several such new approaches are currently being investigated in CAUTIs models, and a few have reached the market, giving hope for solving the problem of resistant-biofilm-related infections [215]. Some of these new, antibiotic-sparing approaches to combating bacterial biofilm-related infections are described below.

### 8.1. Bacteriophages

Bacteriophages, or phages, are viruses highly specific to their host bacteria. They are safe for the human (or animal) patient and bypass classical bacterial resistance mechanisms, making them a promising alternative strategy for fighting biofilm-related infections. Moreover, driven by the decline in new antibiotic development and the growing threat of resistance, phage therapy research has drawn new interest in the last couple of years [216]. Two approaches have been recognized regarding the utilization of phages in combatting biofilms: prevention, which hinders the initiation of biofilm formation, and eradication, which focuses on the elimination of an already established biofilm [217]. For example, UCs coated with a phage cocktail comprising two virulent phages, vB_PmiP_5460 (*Podoviridae* family) and 5461 (*Myoviridae* family), were compared with uncoated controls to assess their ability to inhibit biofilm formation in CAUTI-causing bacteria, including *P. mirabilis*. A significant decrease in biofilm burden was observed on the phage-coated catheters. Notably, the phage cocktail exhibited a pronounced effect of reducing biofilm formation at both 96 and 168 h post-catheterization [218]. These findings are particularly encouraging given the extended indwelling time of UCs in patients. Another phage, SLysin P128, was found to be effective against different strains of staphylococci and managed to suppress their biofilm formation, as demonstrated in an in vivo model of infected rats [219]. Although there are still issues in strain standardization and consistency, phages have proven effective in reducing bacterial contamination on medical catheters, and this success is driving their development as innovative medications specifically targeting infections caused by bacterial biofilms [220].

Moreover, the combination of phages and antibiotics has been suggested as a strategy to address multidrug-resistant (MDR) pathogens and slow down the development of phage resistance [221]. The rationale behind combining these two is that their distinct bactericidal mechanisms are likely to be more effective together than individually. A key example of this is phage–antibiotic synergy (PAS), where the pairing of phages and antibiotics can create a synergistic effect. In addition to enhancing bacterial pathogen suppression, PAS treatment has been shown to reduce biofilm formation and prevent the emergence of resistant strains. For example, a combination of phages with ciprofloxacin or gentamicin was found to successfully eliminate biofilm-embedded *P. aeruginosa* and *S. aureus*, whereas antibiotics or phages alone had only a limited impact [221,222,223]. Furthermore, the combination of phages with antibiotics has been found to re-sensitize bacteria to drugs [224]. For example, PAM2H (phage) restored the sensitivity of *P. aeruginosa* to ceftazidime, ciprofloxacin, gentamicin, and meropenem in vitro and led to a synergistic decrease in bacterial load in vivo [225]. Catheters coated with a phage cocktail targeting *P. mirabilis* demonstrated a marked reduction in *P. mirabilis* biofilm formation within a dynamic biofilm model designed to simulate CAUTIs [226]. Catheters functionalized with phages represent a novel non-antibiotic strategy for the prevention of CAUTIs. These biologically derived agents provide targeted antibacterial effects, interfere with biofilm formation, and mitigate the emergence of antimicrobial resistance. Moreover, incorporating such approaches into catheter design holds potential to enhance infection control, decrease reliance on antibiotics, and support global antimicrobial stewardship initiatives. However, the translation of these innovations into routine clinical practice requires standardized clinical protocols and well-defined regulatory frameworks [227,228].

### 8.2. Antimicrobial Peptides

Antimicrobial peptides (AMPs) represent a diverse class of small peptides found in various organisms, including plants, animals, and microorganisms. They offer a promising antimicrobial-sparing approach, as they exhibit broad-spectrum antimicrobial activity and, when incorporated into medical devices, they can combat biofilms through diverse mechanisms [229]. One of the primary modes of their activity is disrupting the integrity of microbial cell membranes. For example, bacteriocins, namely, nisin A, lacticin Q, and nukacin ISK-1, can disrupt the membrane potential of *S. aureus* cells entrenched within biofilms (including methicillin-resistant *S. aureus* (MRSA) strains). They are typically cationic in nature, and their positive charge allows them to interact with the negatively charged microbial cell membranes [230]. Once bound to the cell membrane, they disrupt membrane integrity, leading to leakage of cellular contents and eventual cell death. Specifically, they inhibit the first adhesion of cells to medical device surfaces. For example, cathelicidin LL-37 disrupts bacterial membranes and hinders attachment, while also downregulating the QS crucial to biofilm development in pathogens like *S. aureus* and *P. aeruginosa* [231]. AMPs exhibit potent activity against multidrug-resistant bacteria, as well as slow-growing or dormant biofilm-forming cells. In contrast to existing antibiotics, AMPs have lower susceptibility to resistance development, thereby offering a promising strategy for biofilm control [232].

### 8.3. Nanoparticles and Other Surface Modification Strategies

Nanoparticles (NPs) show promise in combating biofilm-related infections by offering targeted drug delivery to diseased tissues and an alternative mode of action compared with conventional antimicrobials [233]. Metal and metal oxide nanoparticles, including gold, silver, and copper, exhibit potent antimicrobial properties against bacterial biofilms. The interaction between NPs and biofilms can be viewed as a three-stage process: the movement of NPs towards the biofilm, their attachment to the biofilm surface, and their migration within the biofilm [234]. Moreover, nanoparticles may dissolve in the surrounding fluid (urine in the case of urinary tract) in a pH-dependent manner, releasing metal ions into the environment. For instance, silver nanoparticles (AgNPs) release silver ions (Ag^+^) when they dissolve [235]. These ions can interact with bacterial cell membranes, leading to cell wall disruption, protein denaturation, and DNA interference, ultimately killing the bacteria or inhibiting their growth on different surfaces [236]. Nanoparticles can also induce oxidative stress mechanisms that are effective against biofilm formation on urinary catheters [237]. Oxidative stress occurs when there is a mismatch between the production of reactive oxygen species (ROS) and the ability of biological systems to detoxify these reactive intermediates [238]. By generating ROS, nanoparticles cause extensive damage to bacterial cells and biofilm structures, enhancing the antibacterial efficacy of the catheter surface. This approach not only reduces infection rates but also helps to minimize the development of bacterial resistance, offering a promising strategy for improving urinary catheter design and patient outcomes [239].

CAUTIs may be avoided in large part by applying nanoparticle coatings to UCs [240]. These coatings lower the risk of infection by acting as a barrier to prevent bacterial colonization and biofilm formation on the catheter surface. Through the utilization of nanoparticles, such as copper or silver, these coatings can efficiently impede the growth and proliferation of bacteria on catheters. Furthermore, the surface characteristics of the catheter can be improved by nanoparticles, increasing its resistance to bacterial adhesion and simplifying the removal of bacteria during normal maintenance [241]. Roe and colleagues demonstrated that plastic catheters coated with silver nanoparticles diminish the formation of biofilms and the proliferation of various pathogens, such as *E. coli*, *Enterococcus* spp., *S. aureus*, *P. aeruginosa*, and coagulase-negative Staphylococci, in comparison to catheters lacking this coating [242]. Another specific metallic coating on UCs has been introduced on the market under the name Bactiguard^®^ [243]. This coating minimizes bacterial adhesion to the device surface, effectively preventing biofilm formation, by the induction of a galvanic effect which creates a micro-electrochemical environment that disrupts the bacterial cell wall and inhibits biofilm formation [244].

Impregnation and functionalization with antibiotics or bioactive agents are other strategies to modify catheter surfaces [245,246]. For example, polyurethane catheters functionalized with nitric oxide-releasing copolymers markedly suppressed biofilm formation by *P. aeruginosa* and *S. aureus* [247]. Similarly, ciprofloxacin-loaded hydrogels enabled a sustained release of antibiotics for 7 days, effectively reducing CAUTI incidence and delaying *E. coli*-associated bacteriuria [248]. Bioactive compounds can be incorporated onto catheter surfaces through ionic interactions, as well as covalent or hydrogen bonding with the available functional groups [249]. This approach is particularly advantageous for biomolecules with poor solubility, weak adsorption capacity, or random orientation, conditions that often enhance antimicrobial efficacy and drug stability. However, a major limitation comes from the poor binding efficiency of such bioactive compounds to commonly used catheter materials, such as polyurethane, silicone, and latex, which are highly hydrophobic and lack stable binding capacity [250]. To overcome this limitation, catheter surfaces can be modified through functionalization strategies that introduce reactive chemical groups, thereby enabling stable bonding with bioactive agents. These modifications can be achieved through chemical treatments that introduce positively charged groups (e.g., amino groups via ethylenediamine) or negatively charged groups (e.g., carboxyl groups via carbon dioxide plasma), as well as radiation-based approaches, such as ultraviolet or gamma irradiation, which create reactive sites on the catheter surface [245,249]. The summary of mechanism of action and delivery methods of nanoparticles against biofilm are represented in Table 3.

### 8.4. Phytochemicals

Phytochemicals (secondary plant metabolites) refer to naturally occurring compounds found in plants that may have various beneficial effects on human and animal health, including potent anti-biofilm properties [256]. Phytochemicals can be categorized into approximately eight major classes: flavonoids, terpenoids, lectins, alkaloids, polypeptides, polyacetylenes, phenolics, and essential oils [257]. They exert anti-biofilm effects by employing diverse mechanisms distinct from conventional antibiotics (including inhibition of quorum sensing, motility, adhesion, and the generation of ROS, among other actions) [258]. For instance, the main groups of phenolic compounds are phenolic acids, quinones, flavones, flavonols, tannins, and coumarins [259]. Phenolic compounds have demonstrated antibacterial activity against *S. aureus* and *P. aeruginosa* [260,261]. This suggests that tannins’ characteristic phenolic nature may be a major factor in their antibacterial action [262]. Some plants that are well-known in different countries around the world, such as cranberries and blueberries, are extensively used as medicinal plants for treating UTIs [263]. The antibacterial properties of cranberries and blueberries are believed to be associated with fructose and proanthocyanidins, which inhibit virulence factors like P fimbria by preventing pathogens from colonizing the urinary tract [264]. In some folk medicine traditions, Equisetum debile Roxb and Gomphrena celosioides Mart are considered highly effective and are frequently utilized for treating UTIs [265]. Moreover, antibacterial polyphenols have been found in green tea (Camellia sinensis). In vitro studies have shown that epigallocatechin, a major compound in green tea, has demonstrated antibacterial activity against strains of *E. coli* that cause UTIs. Giri and colleagues (2020) reported in vitro that green tea extracts show considerable potential as antimicrobial agents against drug-resistant UPEC due to the presence of polyphenol compounds, specifically catechins [266]. Numerous in vivo and in vitro studies have demonstrated that extracts from Hydrastis canadensis (goldenseal) exhibit antibacterial activity against Gram-positive uropathogens (including MRSA), which is attributed to alkaloids like berberine, hydrastine, and canadine [267]. Nowadays, combining biomaterials with phytochemicals is an expanding new direction in medical research, particularly for improving the delivery and effectiveness of herbal treatments. This approach seeks to address the natural limitations of herbal extracts, such as low bioavailability and the inability to target specific areas, by using biomaterials as delivery systems [268,269]. 

## 9. Conclusions

The introduction of indwelling medical devices has revolutionized healthcare, significantly improving patient management for diverse medical conditions. However, the vulnerability of these devices, especially UCs, to bacterial colonization and infection remains a pressing issue. The prevalence of CAUTIs highlights the critical need for effective strategies to mitigate their impact on healthcare systems. CAUTIs, representing a significant portion of HAIs, pose severe challenges due to their association with prolonged hospital stays, increased morbidity and mortality, and substantial financial costs. A key factor in the pathogenesis of CAUTIs is the development of bacterial biofilms on catheter surfaces, which hinders standard antimicrobial treatments. The complex interplay in CAUTIs among biofilm-embedded bacteria, the host response, and antimicrobial resistance accentuates the need for innovative treatment and prevention strategies. Combating CAUTIs necessitates an understanding of biofilm formation mechanisms and resistance patterns. Despite the existence of various in vitro models for biofilm research, many operate under static conditions, which limits their efficacy due to rapid nutrient depletion and metabolite build-up. Therefore, advancing dynamic in vitro flow models that more accurately mimic clinical scenarios in UTIs is crucial. These models will replicate the PK profiles of drugs used in UTIs, paving the way for the development of new therapeutic strategies with enhanced clinical effectiveness. This progression is essential to addressing the complex nature of CAUTIs and ultimately improving patient outcomes.

## Figures and Tables

**Figure 1 ijms-26-09193-f001:**
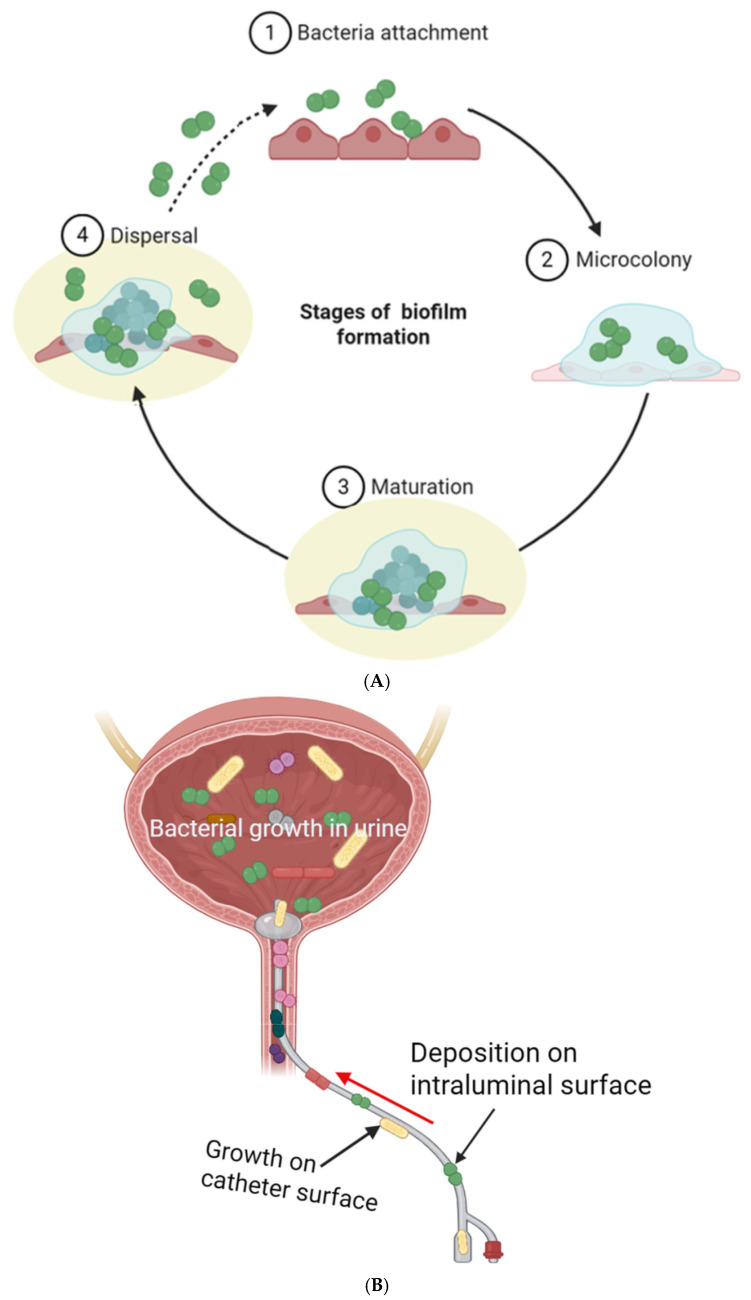
Major stages of biofilm formation (**A**). Mechanism of bacterial biofilm formation in CAUTIs (**B**). Both figures were created with BioRender.com (Version 2025).

**Figure 2 ijms-26-09193-f002:**
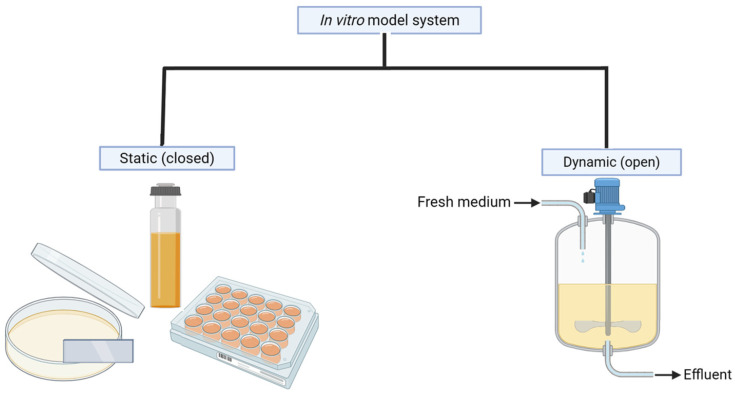
Different types of in vitro models; created using Biorender.com (Version 2025).

**Figure 3 ijms-26-09193-f003:**
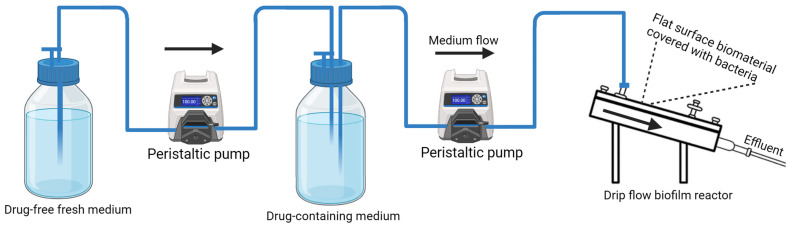
Design of a dynamic in vitro PK/PD model for investigation of drug effects on biofilms; created using Biorender.com. Arrows indicates the medium flow direction from left to right.

**Table 1 ijms-26-09193-t001:** Components of EPSs in biofilms and their roles in biofilm formation and development [45].

Components	Examples of Bacterial Species	Chemical Group	Functions
Polysaccharides	*E. faecalis*	Enterococcal polysaccharide antigen (EPA)	Adhesion, biofilm formation, resistance to antibiotics and phagocytosis, and colonization
	*E. coli*	Polysaccharide intercellular adhesin (PIA)	Adhesion, biofilm formation, resistance to antibiotics and phagocytosis, and colonization
	*S. aureus*	Polysaccharide intercellular adhesin (PIA)	Adhesion, cohesion, scaffolding, stability, and protection against antibiotics
	*P. aeruginosa*	Alginate	Adhesion and protection from environmental factors
	*S. mutans*	Glucans/fructans	Adhesion, cohesion, scaffolding, stability, cell-to-cell binding, acidic microenvironment, protection against antimicrobials, and nutrient source
Proteins	*E. faecalis*	Enterococcal surface protein (ESP) and cell wall-anchored protein	Facilitate primary attachment and build-up of biofilm
	*E. coli*	Poly-beta-1,6-N-acetyl-D-glucosamine synthase	Synthesis of PGA polymer, which helps in biofilm adhesion
	*S. aureus*	*S. aureus* surface protein G (SasG)	Adhesion and cell-to-cell binding
	*P. aeruginosa*	Lectins (LecA/LecB)	Adhesion, cell-to-cell binding, stability, and respiratory epithelial cell toxicity
	*S. mutans*	Dextranase	EPS degradation/remodeling
Nucleic acid (DNA or RNA)	Wide distribution in bacteria	eDNA	Scaffolding, adhesion, cohesion, nutrient source, DNA damage repair, gene transfer, and interaction with other matrix components
Lipids	*S. aureus*	Teichoic and lipoteichoic acids	Adhesion, cohesion, protection, and immune evasion
Lipopolysaccharides	Wide distribution in Gram-negative bacteria	Lipopolysaccharide (endotoxin)	Adhesion, colonization and host invasion, and activation of immune response

**Table 2 ijms-26-09193-t002:** Mechanisms involved in the acquisition and transmission of antimicrobial resistance genes in biofilms.

Mechanism	Description	Examples	References
Conjugation	Transfer of plasmids carrying ARGs through direct cell-to-cell contact	Plasmid pESI-1 transfers vancomycin resistance among *E. faecalis* within biofilms.	[147]
Transformation	Uptake of free-floating DNA containing ARGs by competent bacteria	*P. aeruginosa* within biofilms is capable of naturally taking up and incorporating both genomic and plasmid DNA. Moreover, *A. baumannii* can acquire DNA from other species, and this transformation may contribute to the rise of multidrug-resistant strains.	[151,152,153]
Transduction	Bacteriophages carrying ARGs transduce recipient bacteria within the biofilm	Bacteriophage CTXφ carries and transfers blaCTX-M genes encoding extended-spectrum β-lactamase resistance in *K. pneumoniae* biofilms.	[154]
MGEs	Plasmids and transposons readily move within the biofilm, facilitating HGT of ARGs	Conjugative transposon Tn1549 encodes vancomycin resistance and spreads among *E. faecium* in biofilms.	[155]

ARGs, antibiotic resistance genes; MGEs, mobile genetic elements; HGT, horizontal gene transfer.

**Table 3 ijms-26-09193-t003:** Mechanisms of action and delivery methods of nanoparticles against biofilms.

Nanoparticle Type	Mechanisms of Action	Target Biofilm-Forming Bacteria	Delivery Methods	Reference
Silver nanoparticles (AgNPs)	Membrane disruption, reactive oxygen species (ROS) generation, and protein inactivation	*S. aureus*, *E. coli*, and *P. aeruginosa*	Coatings and topical application	[251]
Zinc oxide nanoparticles (ZnO NPs)	Membrane disruption, ROS generation, and DNA damage	*S. aureus*, *S. enterica*, and *E. coli*	Coatings, hydrogels, and nanoparticles in solution	[252]
Copper nanoparticles (CuNPs)	Membrane disruption, metal ion release, and enzyme inhibition	*E. coli*, *P. aeruginosa*, and *S. aureus*	Surface modification and nanoparticles in solution	[253]
Titanium dioxide nanoparticles (TiO_2_ NPs)	Photocatalytic activity, ROS generation, and membrane disruption	*S. aureus*, *E. coli*, and *P. aeruginosa*	Coatings and nanoparticle–polymer composites	[254]
Gold nanoparticles	Generation of ROS and interference with metabolic processes	*E. coli* and *S. aureus*	Surface modification and nanoparticles in solution	[255]

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
