# Peer review of "Catheter-Associated Urinary Tract Infections: Understanding the Interplay Between Bacterial Biofilm and Antimicrobial Resistance"

_ijms, 2025, doi:10.3390/ijms26189193_

Round 1
Reviewer 1 Report
Comments and Suggestions for Authors
General Assessment
The manuscript presents a well-structured and timely review of catheter-associated urinary tract infections (CAUTIs), with a focus on the dual role of catheters in clinical care and their contribution to biofilm-associated infections. The review is relevant to the field of infectious diseases, microbiology, and antimicrobial resistance. The emphasis on dynamic in vitro bladder models is a strength, as these systems represent a critical step toward bridging laboratory findings with clinical conditions. The manuscript is clear and comprehensive overall, but there are areas where the content could be strengthened, particularly in terms of depth of analysis and completeness of reference coverage.
Major Scientific Content Comments
- Completeness and Relevance of the Review
- The manuscript provides a broad overview of CAUTIs and associated biofilms; additional discussion of recent advances in in vivo infection models could strengthen the review.
- While dynamic in vitro models are appropriately highlighted, there is little discussion of their limitations compared to clinical settings in Section 1. and 7.2 bacterial biofilm models.
- The review identifies the challenge of biofilm-associated antimicrobial resistance but does not sufficiently discuss emerging non-antibiotic strategies (e.g., bacteriophage therapy, antimicrobial peptides, or catheter surface modifications) in the Introduction
2. Identification of Knowledge Gaps
-
- The review would benefit from a clearer articulation of gaps in current For example, how host immune responses interact with biofilm persistence in CAUTIs is not extensively covered in the Introduction section.
- Section 8. Novel strategies to combat bacterial biofilms: The manuscript could benefit from a discussion of strategies for modifying catheter surfaces.
3. Figures
-
- All figures in the manuscript seem to have insufficient resolution. In addition, please eliminate the duplicate figure footnote (Figures 2,3)
4. Appropriateness of References
-
- Many references are relevant and appropriate, but some key recent studies (within the past 3-4 years) on antimicrobial coatings, catheter materials, and phage therapy for CAUTIs appear to be missing.
- References 67, 68, 91, 123, 147, 184, and 219 should be checked carefully for accuracy and formatting consistency.
5. Scientific Accuracy and Clarity (Specific Lines)
-
- Lines 23–24: Please ensure all scientific names ( coli, E. faecalis, P. aeruginosa, P. mirabilis, K. pneumoniae, S. aureus, Candida spp.) are italicized.
- Line 514: in vitro should be
- Lines 699–700: Sentences should be rephrased for clarity:
- Suggested: “In vitro studies have shown that epigallocatechin, a major compound in green tea, has …”
- Suggested: “Giri and colleagues (2020) reported in vitro that green …”
Additional Minor Editorial Comments
-
- Line 82: Please correct to “catheterization” (US English).
- Lines 436, 665: Consider adjusting to US English conventions (check journal style guide for consistency).
- Line 320: Please remove double
- Line 387: Please correct the typographical
- Abbreviations: Consider defining full form before first use (e.g., Line 55 and Line 206).
Overall Recommendation
The manuscript is well-written, relevant, and scientifically robust as a review article. With careful incorporation of recent literature, clearer identification of knowledge gaps, and refinement of certain sections for clarity and balance, it has the potential to make a significant contribution to the field following major revisions.
Strengths
-
- The manuscript addresses a highly relevant clinical problem (CAUTIs) with significant healthcare impact.
- Well-structured and clearly written, making it accessible to a broad
- Highlights dynamic in vitro bladder infection models, an important and emerging area in CAUTI research.
- Provides a comprehensive overview of uropathies and biofilm-associated antimicrobial
- Conclusions are consistent with the evidence and literature
Weaknesses
-
- Limited discussion of alternative or complementary strategies (e.g., phage therapy, antimicrobial peptides, catheter coatings).
- Knowledge gaps are mentioned but not sufficiently emphasized, particularly regarding host-pathogen interactions and immune responses.
- Missing discussion of in vivo models (animal or clinical studies) that complement in vitro
- Figures/tables could be more effectively used to summarize complex information (e.g., uropathogen prevalence, biofilm mechanisms, model comparisons).
- Some recent and relevant references are missing; accuracy/formatting of specific references (67, 68, 91, 123, 147, 184, 219) should be checked.
Author Response
Summary
Thank you very much for taking the time to review this manuscript. We appreciate your insightful comments and valuable suggestions. Please find the detailed responses below and the corresponding revisions and corrections in track changes mode in the re-submitted manuscript.
Major scientific content comments
- Completeness and relevance of the review
Comment 1: The manuscript provides a broad overview of CAUTIs and associated biofilms; additional discussion of recent advances in in vivo infection models could strengthen the review.
Response 1: Thank you for pointing this out and for your insightful comment. We agree with this comment. Accordingly, we have added further discussion to emphasize this point. Additional relevant information regarding to recent advances in in vivo infection models has been incorporated into the manuscript. The revisions can be found on page 17-18, line 667-689 in the “all markup” mode (all changes visible). The updated text is now included in the manuscript. We tried our best to find relevant works on in vivo models to provide a concise summary of the advances in this field, however, if the Reviewer still thinks that we missed some particular and relevant work, we would appreciate pointing it out so we could cover it in this section.
Comment 2: While dynamic in vitro models are appropriately highlighted, there is little discussion of their limitations compared to clinical settings in Section 1. and 7.2 bacterial biofilm models.
Response 2: Again, thank you for your insightful comment, which we agree with. Accordingly, we have added further discussion to emphasize this point. Additional relevant information regarding the limitations of the dynamic in vitro models (including dynamic bladder infection model) as compared to clinical settings has been incorporated into the manuscript. The revisions can be found on page (16) in section 7.1.2. dynamic……, line (575-602). The updated text is now included in the manuscript. The limitations (and some advancements to overcome them) of in vivo CAUTI bacterial biofilm models in regard to clinical settings have been discussed now in Section 7.2. (as discussed in Response 1).
Comment 3: The review identifies the challenge of biofilm-associated antimicrobial resistance but does not sufficiently discuss emerging non-antibiotic strategies (e.g., bacteriophage therapy, antimicrobial peptides, or catheter surface modifications) in the Introduction
Response 3: Thank you for pointing this out. We agree with this comment. Accordingly, additional relevant information concerning emerging non-antibiotic strategies has been incorporated into the introduction. The revisions can be found on page 3, lines 93-105. The updated text is now included in the manuscript.
- Identification of knowledge gaps
Comment 4: The review would benefit from a clearer articulation of gaps in current. For example, how host immune responses interact with biofilm persistence in CAUTIs is not extensively covered in the Introduction section.
Response 4: Again, thank you for pointing this out. We agree with this comment. We have accordingly added further discussion to emphasize this point. Additional relevant information concerning the summary of challenges in CAUTI research (e.g. host immune-biofilm interaction) as well as the perspectives of new treatment options has been incorporated into the manuscript. The revisions can be found on page 3, line 93-106.
Comment 5: Section 8. Novel strategies to combat bacterial biofilms: The manuscript could benefit from a discussion of strategies for modifying catheter surfaces.
Response 5: Again, thank you for pointing this out. We agree with this comment. Accordingly, we have added relevant information on strategies for catheter surface modification, including impregnation and functionalization with antibiotics or bioactive agents, in addition to nanoparticle coatings. The added section can be found on page 20-21, line 805-824. The updated text is now included in the manuscript.
- Figures
Comment 6: All figures in the manuscript seem to have insufficient resolution. In addition, please eliminate the duplicate figure footnote (Figures 2,3)
Response 6: We have adjusted the resolution of all figures, as well as corrected the footnotes for Figures 2 and 3.
- Appropriateness of references
Comment 7: Many references are relevant and appropriate, but some key recent studies (within the past 3-4 years) on antimicrobial coatings, catheter materials, and phage therapy for CAUTIs appear to be missing.
Response 7: We agree with the comments and suggestions. Accordingly, we have incorporated recent references into the aforementioned sections. The updated text is now included in the manuscript. We tried our best to find relevant works on antimicrobial coatings, catheter materials, and phage therapy for CAUTIs to provide a concise summary of the advances in this field, however, if the Reviewer still thinks that we missed some particular and relevant work, we would appreciate pointing it out so we could cover it in this section.
Comment 8: References 67, 68, 91, 123, 147, 184, and 219 should be checked carefully for accuracy and formatting consistency.
Response 8: Thank you for the comments, we have adjusted the formatting in references. The updated text is now included in the manuscript (reference section).
- Scientific accuracy and clarity (specific lines)
Comment 9: Lines 23–24: Please ensure all scientific names (E. coli, E. faecalis, P. aeruginosa, P. mirabilis, K. pneumoniae, S. aureus, Candida spp.) are italicized.
Response 9: Agree. We have accordingly modified. The modification can be found on page 1, line 25-26.
Comment 10: Line 514: in vitro should be
Response 10: Agree. Accordingly, we have italicized the word ‘in vitro’ as well as ‘in vivo’ and bacterial species throughout the Manuscript.
Comment 11: Lines 699–700: Sentences should be rephrased for clarity:
- Suggested: “In vitro studies have shown that epigallocatechin, a major compound in green tea, has …”
- Suggested: “Giri and colleagues (2020) reported in vitro that green …”
Response: 11. Thank you for your kind suggestions. The modification can be found on page 22, line 846-848. The updated text is now included in the manuscript
Additional minor editorial comments
Comment 12. Line 82: Please correct to “catheterization” (US English).
Response 12: We have modified the format and incorporated into the manuscript. The revisions can be found on page 2, line 85.
Comment 13: Lines 436, 665: Consider adjusting to US English conventions (check journal style guide for consistency).
Response 13: We adjusted the spelling to US English, correction of line 436 is located on line 487 and line 665 is located on line 791. The updated text is now included in the manuscript
Comment 14: Line 320: Please remove double
Response 14: Correction can be found on page 10, line 343. The updated text is now included in the manuscript.
Comment 15: Line 387: Please correct the typographical
Response 15: The correction regarding the typography can be found on page 12, line 435. The updated text is now included in the manuscript.
Comment 16: Abbreviations: Consider defining full form before first use (e.g., Line 55 and Line 206).
Response 16: Thank you for pointing this out. We have modified the format and incorporated into the manuscript. The correction regarding the full name of CAUTIs can be found on page 2, line 57 for CAUTIs expanded name and PNAG page 6, line 228. The updated text is now included in the manuscript.
Reviewer 2 Report
Comments and Suggestions for Authors
Review for ijms-3831298
In this review article entitled “Catheter-associated urinary tract infections: Understanding the interplay between bacterial biofilm and antimicrobial resistance”, the authors (Tegegne et al.,) reviewed catheter-associated urinary tract infections (CAUTIs), explored the key role of biofilms in the responses to antimicrobial treatments. The authors highlighted also advances in several model systems and presented some promising new approaches to combat CAUTIs.
Overall, the manuscript is acceptable but requires some improvements. Hereafter some comments revealed after reviewing its current version.
- It is recommended to add some limitations of the study as this would have additional value to the manuscript, particularly enriching the discussion.
- The reviewing approach is consistent, appropriate and technically sound.
- The physiological heterogeneity and reduced metabolic activity have considerable interest in the reviewed subject but in the manuscript it is not well discussed.
- Some figures (2 and 3) have duplicate titles; in the figure themselves and the figures legends. Please check and correct accordingly.
- The sentences “It is estimated that …and animals”, “The ability of biofilm…associated with UCs” and “As a result, these …related infections” can be supported by the following relevant and recent reference doi: 10.1155/cjid/2333207.
- English language is overall acceptable just some minor editing and checking are required.
Minor comment
- Use italic for the species such as “P. aeruginosa”, “E. coli” throughout the whole manuscript not only in table 1
Author Response
Summary
Thank you very much for taking the time to review this manuscript and for insightful comments. Please find the detailed responses below and the corresponding revisions/corrections in the track changes mode in the re-submitted files.
Comment 1: The physiological heterogeneity and reduced metabolic activity have considerable interest in the reviewed subject but in the manuscript, it is not well discussed.
Response 1: Thank you for pointing this out. We agree with this comment. We have accordingly added further discussion to emphasize this point. Additional relevant information concerning physiological heterogeneity and reduced metabolic activity has been incorporated into the manuscript. The revisions can be found on page 11, line 390-414 in the “all markup” mode (all changes visible). The updated text is now included in the manuscript.
Comment 2: Some figures (2 and 3) have duplicate titles; in the figure themselves and the figures legends. Please check and correct accordingly.
Response 2: Thank you for pointing this out. We agree with this comment. We have corrected the title of the figures and resolution for all figures.
The correction can be found on page 14 in section 7.1. In vitro biofilm models line 512 for figure 2.
The correction can be found on page 16 in section 7.1.2. Dynamic system, line 603 for figure 3.
Comment 3: The sentences “It is estimated that …and animals”, “The ability of biofilm…associated with UCs” and “As a result, these …related infections” can be supported by the following relevant and recent reference doi: 10.1155/cjid/2333207.
Response 3: We agree with this comment. Accordingly, we have added this relevant and recent reference into the manuscript. The correction can be found on page 3, line 93.
Comment 4: English language is overall acceptable just some minor editing and checking are required.
Response 4: Thank you, we have checked the manuscript again and corrected the typos we found.
Comment 5: Use italic for the species such as “P. aeruginosa”, “E. coli” throughout the whole manuscript not only in table 1
Response 5: Agree, we italicized all bacterial species in the whole manuscript.